# Adipocyte adaptive immunity mediates diet-induced adipose inflammation and insulin resistance by decreasing adipose Treg cells

Tuo Deng[1,2], Joey Liu[3], Yanru Deng[1], Laurie Minze[4], Xiang Xiao[4], Valerie Wright[3], Richeng Yu[1,5], Xian C. Li[4], Alecia Blaszczak[3,7], Stephen Bergin[6,7], David DiSilvestro[3], Ryan Judd[7], David Bradley[3], Michael Caligiuri[7], Christopher J. Lyon[1,†] & Willa A. Hsueh[3]

Obesity leads to a switch in subsets of CD4$^+$ T cell in adipose tissue, characterized by an increase in IFN$\gamma$ producing Th1 cells and a decrease in anti-inflammatory regulatory T (Treg) cells, which impairs systemic insulin sensitivity. What signals these changes is unknown. Herein we demonstrate that genetic deficiency of adipocyte MHCII decreases adipose IFN$\gamma$ expression and increases adipose Treg abundance in obese mice, leading to reduced obesity-induced adipose inflammation and reduced insulin resistance without affecting weight gain. The preserved insulin sensitivity of high fat diet (HFD)-fed adipocyte-specific MHCII knockout (aMHCII$^{-/-}$) mice was substantially attenuated by adipose-specific Treg ablation. Adipocytes of aMHCII$^{-/-}$ mice exhibit decreased capacity to stimulate IFN$\gamma$ production in Th1 cells, whereas HFD-fed IFN$\gamma$R1$^{-/-}$ mice were more insulin sensitive and had similarly high levels of Tregs in adipose tissue as aMHCII$^{-/-}$ mice. We further show that IFN$\gamma$ strongly inhibits IL-33 effects to promote adipose Treg proliferation. Our results identify MHCII in adipocyte as a critical determinant of the obesity-induced adipose T cell subset switch and insulin resistance.

[1] Center for Bioenergetics, Houston Methodist Research Institute, Weill Cornell Medical College, Houston 77030, Texas, USA. [2] Diabetes Center, Institute of Metabolism and Endocrinology, The Second Xiangya Hospital and Key Laboratory of Diabetes Immunology (Central South University), Ministry of Education, Changsha 410011, China. [3] The Diabetes and Metabolism Research Center, Department of Internal Medicine, Division of Endocrinology, Diabetes and Metabolism, The Ohio State University, Columbus, Ohio 43210, USA. [4] Immunobiology and Transplantation Research, Houston Methodist Research Institute, Texas Medical Center, Houston, Texas 77030, USA. [5] Guizhou Provincial People's Hospital, Guiyang 550002, China. [6] The Comprehensive Cancer Center–Arthur G. James Cancer Hospital and Richard J. Solove Research Institute, The Ohio State University, Columbus, Ohio 43210, USA. [7] Medical Scientist Training Program and Biomedical Sciences Graduate Program, The Ohio State University, Columbus, Ohio 43210, USA. [†] Present address: Virginia G. Piper Biodesign Center for Personalized Diagnostics, The Biodesign Institute, Arizona State University, Tempe, Arizona 85287, USA. Correspondence and requests for materials should be addressed to T.D. (email: Tdeng@houstonmethodist.org) or to W.A.H. (email: Willa.Hsueh@osumc.edu).

Obesity is a world-wide epidemic with multiple systemic inflammatory-induced complications, including insulin resistance (IR), a major pathophysiologic change that is implicated in metabolic syndrome and type 2 diabetes mellitus[1]. Understanding the nature of the adipose inflammation could be important for developing better therapies to combat these complications. Mounting evidence indicates that characteristic changes in adipose-resident T cell (ART) quantity and polarization during excess weight gain are key determinants of systemic insulin action. CD4$^+$ ARTs are among the first adipose-resident immune cells to change in response to HFD challenge[2]. HFD induces a rapid increase in the number of pro-inflammatory Th1 CD4$^+$ ARTs, followed by a decrease in CD4$^+$ adipose Tregs that attenuate adipose inflammation and systemic IR[3–5]. Reducing the HFD-induced Th1-to-Treg ART ratio by anti-CD3 antibody injection or by adoptive transfer of CD4$^+$ T cells into Rag1-deficient mice[4] or immunotherapeutic approaches that increase adipose Tregs in HFD-fed obese mice[3–6] prevents obesity-induced IR, emphasizing the role of adipose Tregs in regulating systemic insulin sensitivity. Similarly, mice that do not express Tbet, which directs Th1 lineage commitment and IFNγ expression, are more insulin sensitive than wild-type (WT) mice when challenged with HFD, despite gaining more weight[7]. Tbet null mice also have more adipose Tregs than WT mice, and this increase is strongly implicated in their improved metabolic profile[7,8]. Despite these studies, little is known about the specific mechanisms that activate pro-inflammatory T cell changes in obesity.

Adipocytes function as both storage and endocrine cells and are thus uniquely poised to sense and respond to caloric excess by expressing signals that initiate adipose inflammation. Adipocytes comprise the bulk of adipose tissue, but their role is often overlooked in studies of adipose immune responses even though they secrete multiple hormones and cytokines that regulate immune cell behaviour, including leptin, resistin, adiponectin and others[9]. Adipocytes also express MHCII molecules[10,11], a series of proteins that control presentation of protein antigens to CD4$^+$ T cells. In vitro cultured primary adipocytes are capable of activating CD4$^+$ Th1 cells[10]. Adipocytes can also present lipid antigens via CD1d to modulate invariant natural killer T (iNKT) cell function[12]; this interaction activates iNKT cells to prevent adipose tissue inflammation and IR in the lean state, but is substantially down-regulated in obesity[13]. However, the contribution of adipocyte antigen presentation as a determinant of the adipose T cell repertoire is not known.

We therefore generated mice with adipocyte-specific MHCII deficiency (aMHCII$^{-/-}$) to address this question, and now report that aMHCII$^{-/-}$ mice are protected from diet-induced CD4$^+$ ART changes, adipose inflammation and systemic IR, indicating that the adipocyte is an important regulator of these obesity-induced phenotypes. Furthermore, we show that the mechanism of protection involves a marked increase in adipose Tregs resulting from decreased production of IFNγ. These changes were associated with decreased adipocyte inflammatory gene expression and macrophage metabolic activation. Thus, adipocyte MHCII may be a target to prevent adipose inflammation and its metabolic complications.

## Results

### Generation of mice with adipocyte-specific MHCII deficiency.
C57BL/6 mice express only MHCII H2-A[14], a heterodimer of H2-Aα and H2-Aβ1 subunits, and H2-Ab1-deficient C57BL/6 mice cannot form functional MHCII complexes to activate CD4$^+$ T cells[15]. We therefore crossed C57BL/6-background H2-Ab1$^{flox/flox}$ (WT) mice with Adipoq-cre mice to create adipocyte-specific MHCII knockout (aMHCII$^{-/-}$, KO) mice (Fig. 1a,b), which were viable and born at expected Mendelian ratios. Visceral (VAT) but not subcutaneous white adipose tissue (SAT) of aMHCII$^{-/-}$ mice fed a standard chow diet revealed a marked decrease in H2-Ab1 expression not found in other tissues (Fig. 1c). Notably, epidydimal white adipose tissue (eWAT) has higher H2-Ab1 messenger RNA (mRNA) levels than inguinal white adipose tissue (iWAT) (Supplementary Fig. 1). H2-Ab1 expression was similarly reduced in adipocytes but not stromal vascular fractions (SVFs), bone marrow derived macrophages (BMDMs), peritoneal macrophages (PMs) or spleen dendritic cells (DCs) of aMHCII$^{-/-}$ mice (Fig. 1d), while MHCI gene expression did not differ (not shown). H2-Ab1 protein level was decreased in aMHCII$^{/-}$ versus WT mouse adipocytes, which notably exhibited reduced capacity to act as antigen-presenting cells (APCs, Fig. 1e,f). SVF H2-Ab1 levels did not differ. No differences in weight gain, body composition, insulin-sensitivity (Fig. 1g,h) or fasting plasma glucose and insulin (not shown) were observed between chow-fed WT and aMHCII$^{-/-}$ mice, however, indicating that adipocyte-MHCII-mediated CD4$^+$ ART activation did not affect systemic metabolism in lean mice. Flow analyses of SVF revealed no differences in CD4$^+$ or CD8$^+$ T cells as a % of CD45$^+$ cells, and no difference in Tregs, as a %CD4$^+$ cells in aMHCII$^{-/-}$ versus WT mice (Fig. 1i).

### Adipocyte MHCII expression promotes HFD-induced IR.
Adipocyte MHCII expression increases within 2–3 weeks after HFD challenge, corresponding to pro-inflammatory changes in CD4$^+$ ART markers[10], and preceding an M1 adipose tissue macrophage (ATM) accumulation that occurs after about 10–12 weeks HFD exposure. We thus hypothesize that adipocyte MHCII contributes to early CD4$^+$ Th1 ART accumulation during weight gain. Therefore, WT and aMHCII$^{-/-}$ littermate mice were fed HFD and analysed for differences in adipose inflammation and IR after 6 and 12 weeks HFD, corresponding to intermediate and late responses to HFD challenge. WT and aMHCII$^{-/-}$ mice revealed equivalent weight and body composition changes when fed HFD, but aMHCII$^{-/-}$ mice were significantly more insulin sensitive and glucose tolerant than their WT littermates, and while fasting plasma glucose did not differ by genotype, fasting plasma insulin and HOMA-IR were differentially reduced with time on HFD in aMHCII$^{-/-}$ mice (Supplementary Fig. 2a–c and Fig. 2a–d). Western blotting of whole fat and skeletal muscle revealed greater insulin responsiveness assessed by post-insulin AKT phosphorylation (Fig. 2e).

### Adipocyte MHCII enhances adipose inflammation during HFD challenge.
Adipose expression of ART (Cd3) and ATM (Emr1) marker genes did not differ by genotype at either 6 or 12 weeks HFD (Supplementary Fig. 2d and Fig. 2f). However, by 12 weeks HFD, aMHCII$^{-/-}$ mice had fewer crown-like structures (CLS; Fig. 2g), which form due to macrophage accumulation around dead and dying adipocytes and correlate with adipose inflammation[16,17]. No differences were detected in adipocyte size (Fig. 2h,i) at 12 weeks HFD, suggesting that the differential CLS phenotype was not due to differences in hypertrophy-induced adipocyte death.

Consistent with decreased adipose inflammation in HFD-fed aMHCII$^{-/-}$ versus WT mice, adipocyte expression of multiple MHCII-related genes (Ciita, H2-Eb1, Cd74, Cd80 and Cd86) and pro-inflammatory factors (Tnf, Nos2, Il1b and Nlrp3) was decreased or showed trends to decrease at both intervals, as did adipocyte expression of Il10, a primarily anti-inflammatory factor which increases with adipose inflammation (Supplementary Fig. 2e

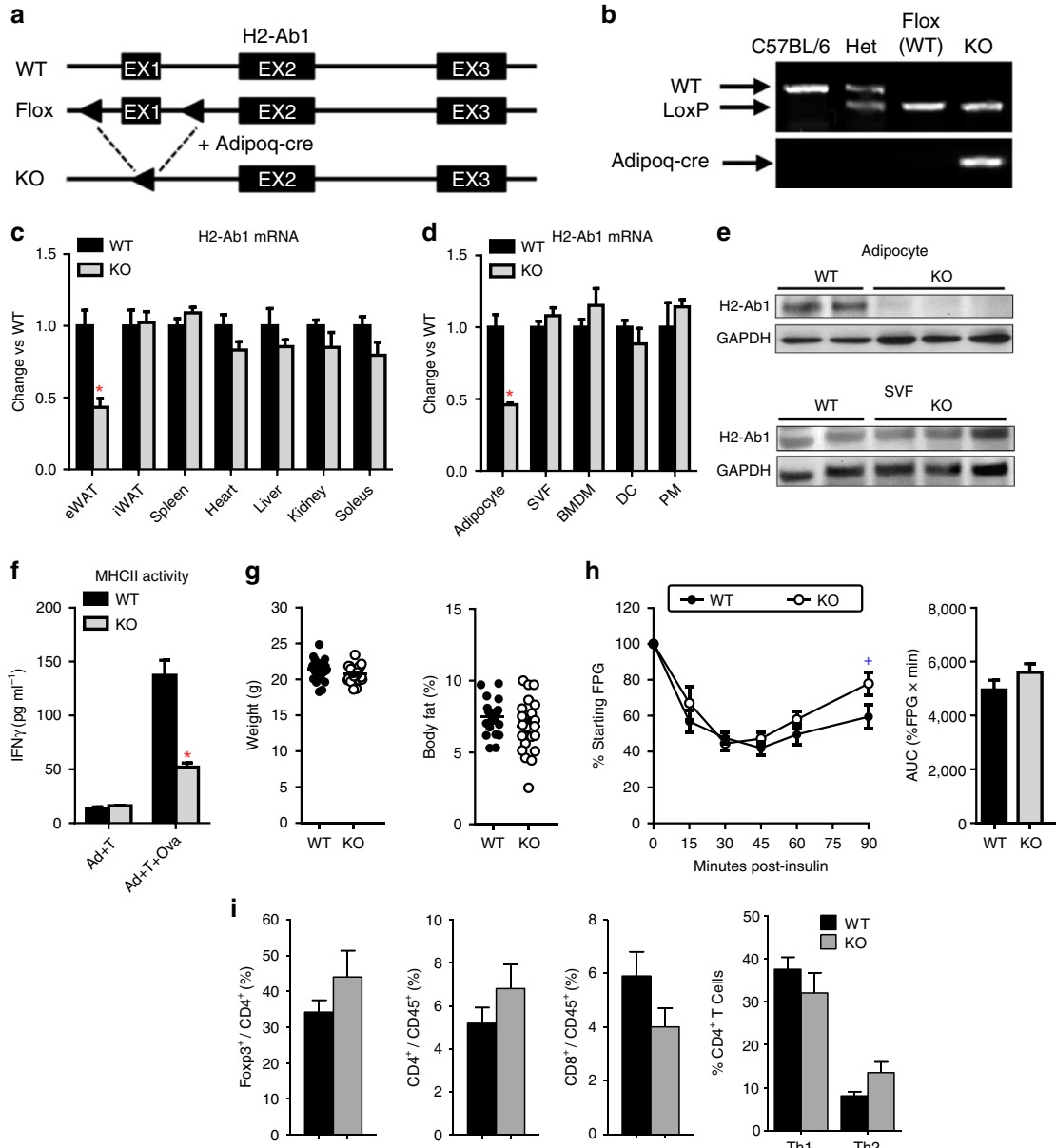

**Figure 1 | Chow-fed aMHCII$^{-/-}$ mice reveal adipocyte-specific MHCII deficiency.** (**a**) Adipocyte-specific gene knockout strategy where triangles designate LoxP sites flanking exon1 (EX1) of H2-Ab1. (**b**) PCR genotyping approach. H2-Ab1 mRNA expression in **c** tissues (N = 6 per group) and (**d**) cell types (adipocytes, SVF, BMDMs, DCs and PMs, see text) of wild-type (WT; H2-Ab1$^{fl/fl}$) and aMHCII$^{-/-}$ (KO) mice fed standard chow diet (N = 4 per group). (**e**) H2-Ab1 protein expression in adipocytes and SVF of WT and KO mice fed HFD measured by Western blot. (**f**) MHCII APC activity of highly-purified adipocytes from chow-fed WT and aMHCII$^{-/-}$ mice (N = 3 per group). (**g**) Body mass and adiposity (N = 22–23 per group) and (**h**) intraperitoneal insulin tolerance test and area under the curve data (N = 10–14 per group) of chow-fed WT and aMHCII$^{-/-}$ mice. (**i**) Flow cytometry analysis of CD4$^+$, CD8$^+$, Treg$^+$ and Th1, and Th2 in eWAT SVF from chow-fed WT and aMHCII$^{-/-}$ mice (N = 3–4 per group). Mean ± s.e.m.; *$P < 0.05$ or $^+P < 0.1$ versus WT by t-test.

and Fig. 3a). *Adiponectin*, the most highly expressed anti-inflammatory adipokine, was differentially increased in aMHCII$^{-/-}$ adipocytes after 12 weeks of HFD, and *Pparγ*, a key regulator of adipocyte differentiation and metabolism trended to increase in aMHCII$^{-/-}$ mice (Fig. 3a). All of these changes were consistent with reduced adipocyte inflammation.

ART CD4 expression did not differ by genotype, but ARTs of HFD-fed aMHCII$^{-/-}$ mice expressed less IFNγ, Tbet and Cd8 and more Foxp3, consistent with an attenuated pro-inflammatory phenotype resulting from fewer Th1 and CD8$^+$ T cells and more Tregs (Figs 2g and 3b). Notably, the Foxp3 increase in aMHCII$^{-/-}$ mice was VAT-specific, as Foxp3 expression did

not differ in SAT, spleen, liver, heart or skeletal muscle of these mice (Supplementary Fig. 3a).

HFD-fed aMHCII$^{-/-}$ mice also exhibited ATM phenotypes consistent with reduced adipose inflammation. After 6 weeks HFD, ATMs of aMHCII$^{-/-}$ versus WT mice revealed greater expression of the anti-inflammatory M2 marker *Chil3/Ym1* and reduced expression of the pro-inflammatory M1 marker *Itgax/CD11c*, although no differences were detected in other M2 (*Arg1*, *Mrc1* and *Retnla/Fizz1*) and M1 (*TNFα*, *Nos2*, *Il12*) markers (Supplementary Fig. 2g). At 12 weeks HFD, ATMs of aMHCII$^{-/-}$ mice expressed or tended to express more of the M2 markers *Arg1* and *Retnla/Fizz1*, respectively, with a

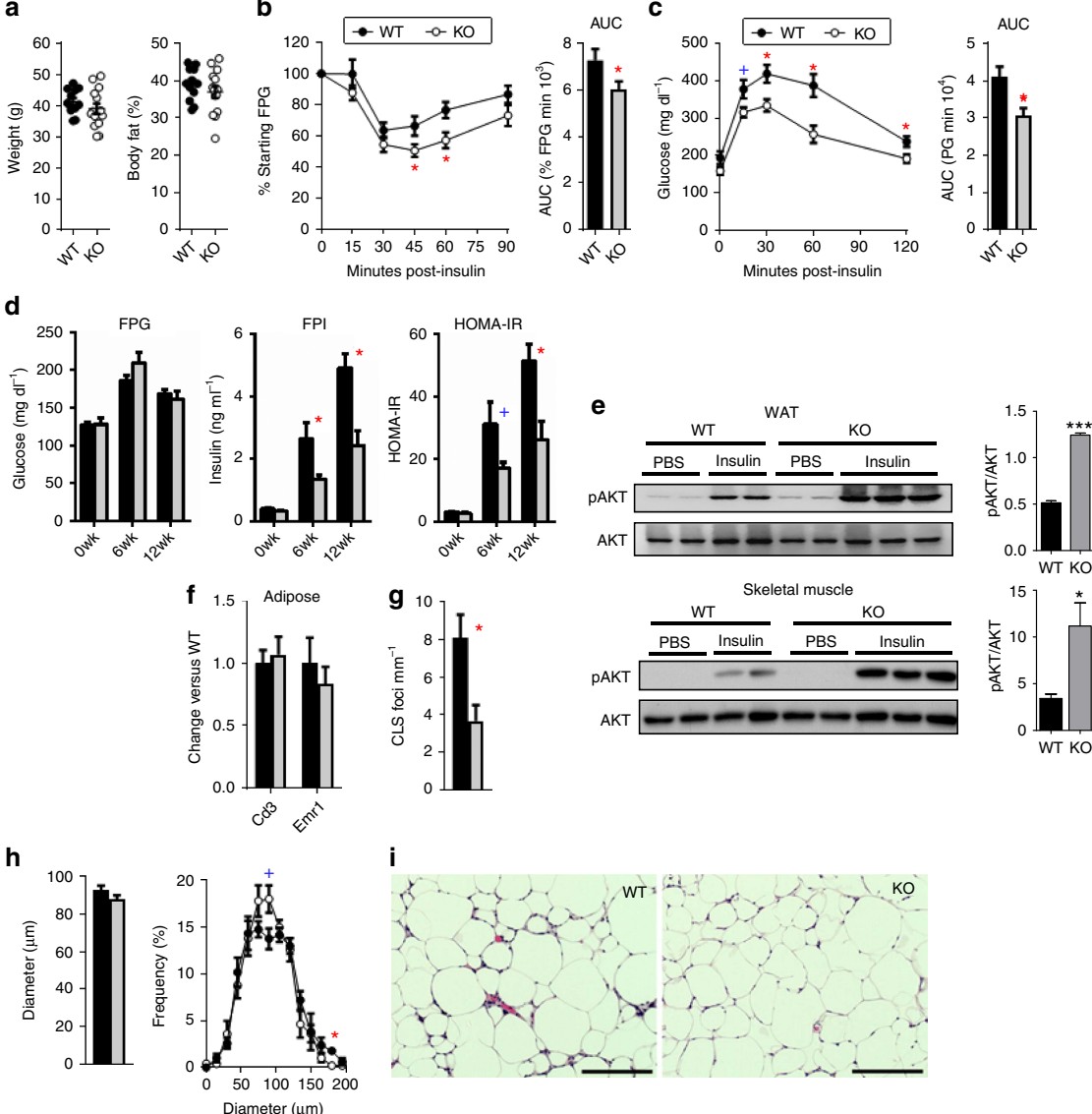

**Figure 2 | Adipocyte MHCII deficiency attenuates insulin resistance. (a)** Body mass and adiposity and **(b)** intraperitoneal insulin tolerance test and area under the curve data and **(c)** intraperitoneal glucose tolerance test and area under the curve data ($N = 11$–14 per group) of 12 week HFD-fed WT and aMHCII$^{-/-}$ mice. **(d)** Fasting plasma glucose, insulin and HOMA-IR values at 0, 6 and 12 weeks of HFD ($N = 6$–18 per group). **(e)** Immunoblots of lysates from eWAT (up panel) and skeletal muscle (down panel) with anti-pAKT and anti-AKT and quantitation of phosphorylated AKT normalized to total AKT. Epididymal adipose tissue **(f)** gene expression, **(g)** crown-like structure (CLS) density, **(h)** adipocyte diameter and distribution ($N = 5$–6 per group) and **(i)** representative adipose tissue histology (scale bar indicates 200 μm) of mice ($N = 5$–6 per group) after 12 weeks HFD. Mean ± s.e.m.; ***$P < 0.001$, *$P < 0.05$ or $^+P < 0.1$ versus WT by $t$-test. CLS values represent data from 25 fields per mouse, while adipocyte diameter values are derived from 100 adipocytes per mouse.

non-significant increase in *Chil3/Ym1* (Fig. 3c), while expression of the M1 markers, *Il12* and *Itgax/Cd11c*, but not *Tnfα* decreased. ATMs may also undergo pro-inflammatory metabolic activation rather than classical M1 activation in response to elevated glucose, insulin and free fatty acids levels during obesity[18]; proposed markers for this metabolic activation phenotype (Abca1, Cd36 and Plin2) were all decreased in ATMs of aMHCII$^{-/-}$ versus WT mice after 12 weeks HFD (Fig. 3c).

**Adipocyte MHCII decreases adipose Treg abundance in HFD-fed mice.** Subsequent flow cytometry studies detected only modest (∼25%) ART and ATM decreases in aMHCII$^{-/-}$ mice after 12 weeks HFD, with no differences in CD4$^+$ or CD8$^+$ ART

abundance (Fig. 3d,e). Adipose Tregs were increased in aMHCII$^{-/-}$ mice at 6 weeks (Supplementary Fig. 2f,h), which was more pronounced at 12 weeks HFD (Fig. 3f,g), consistent with elevated ART Foxp3 mRNA expression and accounting for ∼50% of the CD4$^+$ ART population, similar to levels reported in Chow-fed C57BL/6 mice[19], while conventional CD4$^+$ T cell (Tconv) abundance did not differ (Fig. 3f). Splenic Treg abundance did not differ between aMHCII$^{-/-}$ and WT mice, however, indicating that the increase in adipose Tregs was not a systemic effect (Supplementary Fig. 3b). IL-33 is reported to play a critical role in adipose Treg homoeostasis[20], and adipose Treg expression of the IL-33 receptor ST2 was markedly increased in aMHCII$^{-/-}$ mice (Supplementary Fig. 2h and Fig. 3h), while whole adipose IL-33 mRNA expression did not change,

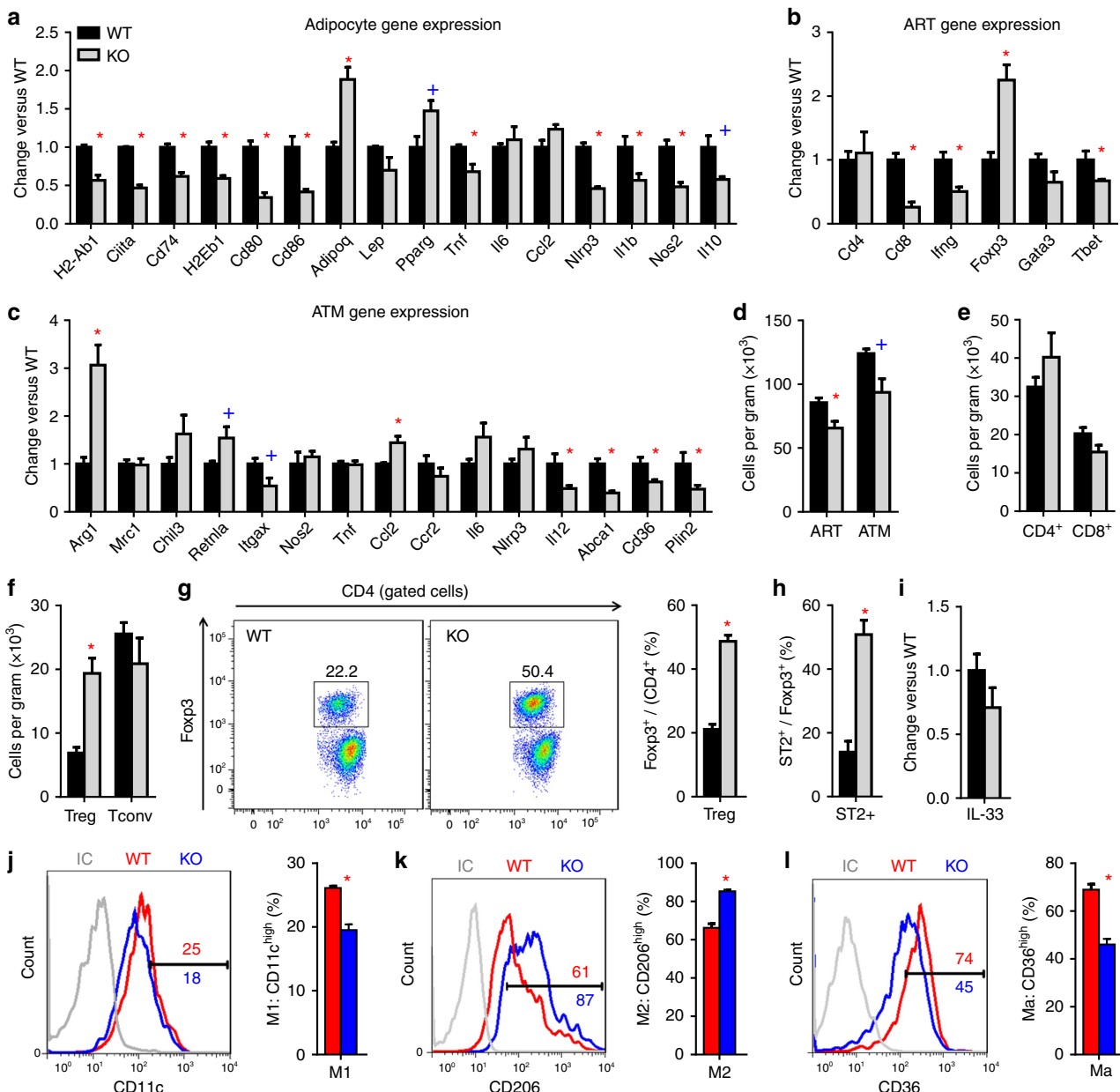

**Figure 3 | Adipocyte MHCII deficiency attenuates adipose inflammation.** Gene expression profiles in **a** adipocytes (**b**) CD3⁺ ARTs and (**c**) ATMs isolated from epididymal adipose tissue at of WT and aMHCII⁻/⁻ (KO) mice at 12 weeks HFD (N = 3–4 per group). Flow cytometry analysis of epididymal adipose tissue (**d**) ART and ATM, (**e**) CD4⁺ and CD8⁺ ART, and (**f**) CD4⁺ Treg and Tconv abundance and (**g**) Treg and (**h**) ST2⁺ Treg prevalence (N = 3 per group). (**i**) Adipose tissue IL-33 expression (N = 5 per group). Flow cytometry analysis of (**j**) CD11c, (**k**) CD206 and (**l**) CD36 in ATMs from HFD-fed WT and KO mice. Ma: Metabolically activated macrophage. Mean ± s.e.m.; *P < 0.05 or ⁺P < 0.1 versus WT by t-test.

suggesting that ST2 expression may account for adipose Treg preservation in HFD-fed aMHCII⁻/⁻ mice (Fig. 3i). ATM flow cytometry data also corresponded with mRNA data, with aMHCII⁻/⁻ mice exhibiting more ATMs positive for the M2 marker CD206, and fewer ATMs positive for the M1 marker (CD11c) and metabolic activation (CD36) (Fig. 3c,j–l). However, these macrophage differences were not present at 6 weeks HFD (Supplementary Fig. 2i).

**IFNγ diminishes insulin sensitivity through changes in adipose Tregs.** Adipose Treg differences are observed at 6 weeks of HFD challenge, before a consistent difference in ATM polarization, implying that adipose Treg responses may be primarily inhibited

by adipocyte- or ART-derived factors. Adipocytes of HFD-fed aMHCII⁻/⁻ mice exhibited reduced capacity to stimulate antigen-specific CD4⁺ activation, as demonstrated by decreased secretion of IFNγ, a major Th1 cytokine (Fig. 4a). IFNγ dose-dependently inhibited Treg differentiation (Fig. 4b) without reducing CD4⁺ T cell numbers (Supplementary Fig. 4). To test the relative contributions of IFNγ versus other putative factors, conditioned media from adipocyte APC assays was added to Treg differentiation cultures in the presence or absence of IFNγ-blocking antibody. Treg differentiation was inhibited by conditioned media from WT but not aMHCII⁻/⁻ adipocyte APC assays. This effect was completely attenuated when Treg differentiation cultures were supplemented with IFNγ-blocking antibody, indicating that IFNγ released in WT adipocyte APC assays

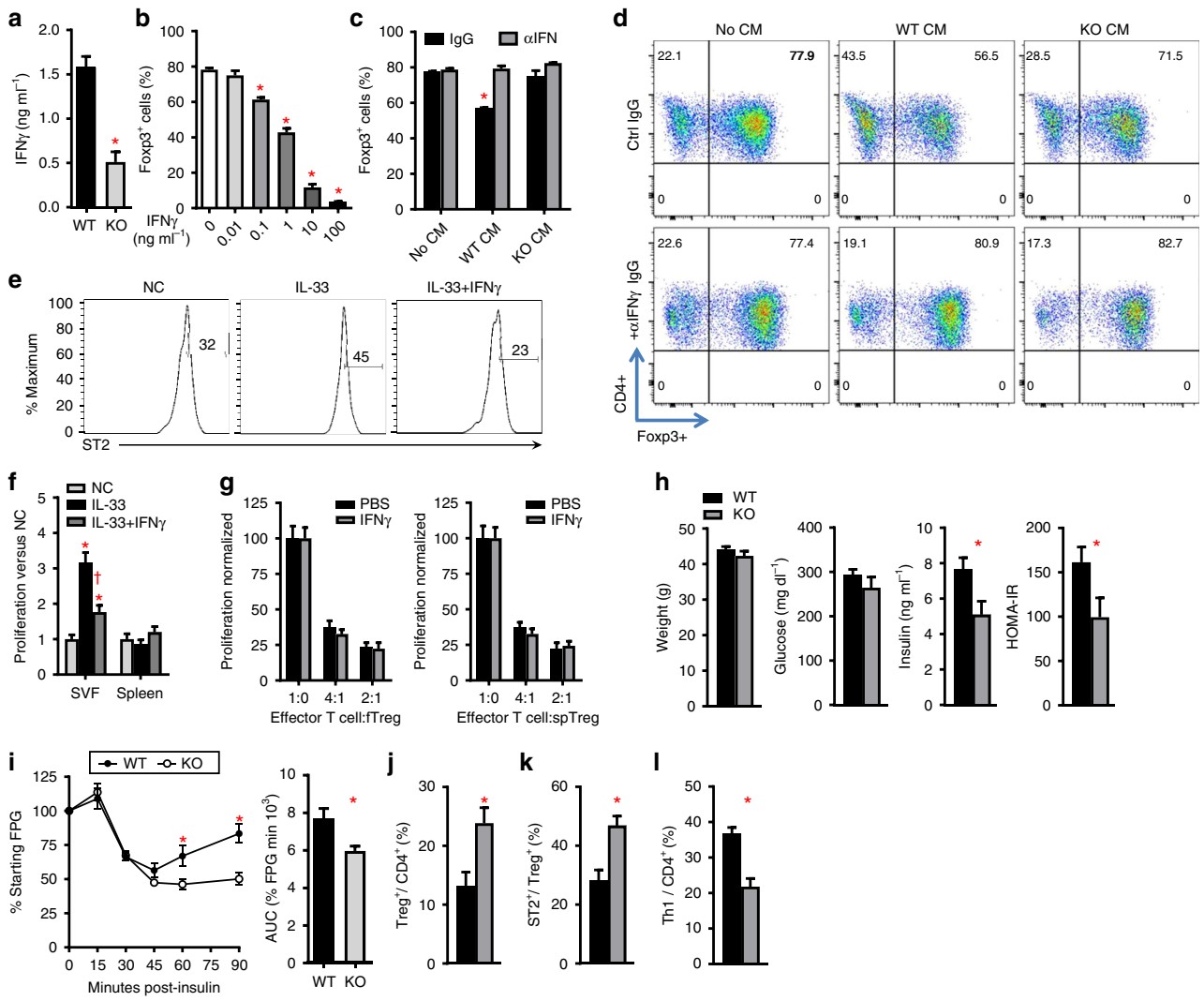

**Figure 4 | Adipocyte MHCII-stimulated IFNγ secretion inhibits Treg differentiation.** (**a**) IFNγ secretion by CD4$^+$ splenic T cells cultured with adipocytes of WT and aMHCII$^{-/-}$ mice and specific antigen. (**b**) Treg differentiation in naïve splenic T cell cultures supplemented with recombinant IFNγ or (**c,d**) 50% conditioned media (CM) from pooled WT or KO cultures (panel A) with IFNγ blocking antibody (αIFN) or non-specific isotype control antibody (IgG). Wild-type VAT (SVF) and spleen lymphocytes were treated with IL-33 or IL-33 plus IFNγ for 3 days. (**e**) Representative histograms of ST2 expression on adipose Tregs after 4 days culture with or without (NC) IL-33 ± IFNγ treatment. (**f**) Graph shows mean fold increase of Treg cells in lymphocytes from SVF or spleen at the end of the culture period. (**g**) 3H-thymidine incorporation in splenic CD4$^+$ effector T cell cultures incubated with adipose (**f**) or splenic (sp) Tregs ± IFNγ. (**h**) Body weight, fasting plasma glucose, insulin and HOMA-IR for 12 week HFD-fed WT and IFNγR1$^{-/-}$ (KO) mice (**i**) Intraperitoneal insulin tolerance test at 12 weeks of HFD. Flow cytometry analysis of epididymal adipose tissue (**j**) Treg to CD4$^+$, (**k**) ST2$^+$ to Treg, (**l**) and Th1 (Tbet$^+$GATA3$^-$) to CD4$^+$ ratios for WT and KO (Mean ± s.e.m.; **a,b**: $N = 4$ per group, **c–g**: $N = 3$ per group, **h**: $N = 10$ (KO), 11 (WT), **i–l**: $N = 11$(KO), 12 (WT), $^†P < 0.05$ versus IL-33 treated group, $^*P < 0.05$ versus all other groups).

was primarily responsible for inhibiting Treg differentiation (Fig. 4c,d).

IL-33 regulates the differentiation and growth of adipose, but not splenic Tregs[20–22] and IL-33 treatment can prevent adipose-specific Treg decreases in HFD-fed mice[23]. Obesity is associated with an increase in adipose IL-33 (refs 22,24), implying that adipose Tregs develop IL-33 resistance during HFD challenge. On the basis of our flow studies, we hypothesized that increased expression of IFNγ by Th1 ARTs might inhibit adipose Treg homoeostasis by decreasing their ST2 expression. We found that IFNγ treatment blocked IL-33-induced increases in proliferation and ST2 expression by adipose but not splenic Tregs (Fig. 4e,f), but had no effect to inhibit the ability of adipose Tregs or splenic Tregs to attenuate antigen-induced CD4$^+$ effector T cell proliferation (Fig. 4g).

These results led us to directly investigate *in vivo* effects of IFNγ in obesity. IFNγR$^{-/-}$ mice challenged with HFD were found to be more insulin sensitive than WT mice after 3 months HFD, despite demonstrating similar weight gain (Fig. 4h,i). Fasting glucose was not different, but fasting insulin and HOMA-IR were less in IFNγR$^{-/-}$ versus WT mice (Fig. 4h). The KO mice had more total and ST2$^+$ adipose Tregs and less adipose Th1 cells (Fig. 4j–l).

**Treg ablation attenuates the improved insulin sensitivity of HFD-fed aMHCII$^{-/-}$ mice.** To test whether preservation of adipose Tregs in HFD-fed aMHCII$^{-/-}$ mice was responsible for their reduced IR, WT and aMHCII$^{-/-}$ mice were immunodepleted and injected with bone marrow isolated from

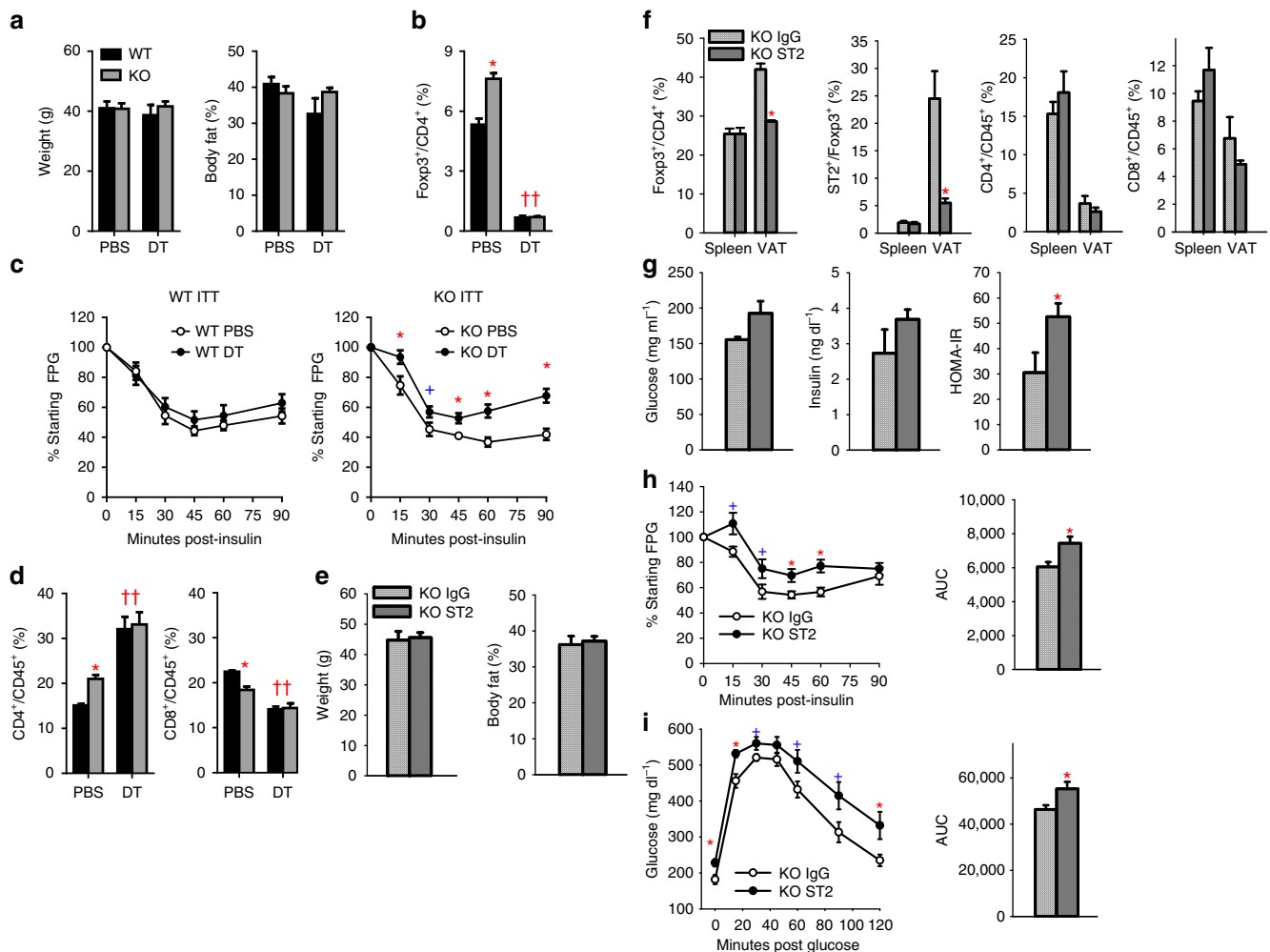

**Figure 5 | Treg ablation during weeks 10–14 of HFD normalizes WT and aMHCII KO mouse phenotypes.** (**a**) Mouse body mass and adiposity, (**b**) adipose Treg prevalence, (**c**) intraperitoneal insulin tolerance test data and (**d**) ART subtype abundance in HFD-fed mice after 4 weeks of PBS or DT treatment. The flow cytometry gating approach is indicated in Supplementary Fig. 8. (**e**) Body mass and adiposity, (**f**) T cell subtype abundance in spleen and VAT, (**g**) fasting blood glucose, fasting plasma insulin, and HOMA-IR, (**h**) intraperitoneal insulin tolerance test and area under the curve data and (**i**) intraperitoneal glucose tolerance test and area under the curve data of >12 week HFD-fed IgG-treated and ST2 antibody-treated aMHCII$^{-/-}$ mice. Mean ± s.e.m.; **a**,**c**: N = 6–9 per group, *$P < 0.05$ or $^{+}P < 0.1$ versus PBS; **b**,**d**: N = 3–4 per group; **e**–**i**: N = 5–7 $P < 0.05$ versus WT (*) or genotype-matched PBS (†).

mice with Treg-specific expression of the human diphtheria toxin receptor. Recipient mice were fed 10 weeks HFD, split into four cohorts with similar body weight and adiposity (Supplementary Fig. 5) and then were maintained on HFD and IP-injected twice-weekly for 4 weeks with PBS or diphtheria toxin. All four groups demonstrated weight and body composition changes (Fig. 5a) that resembled those of non-chimeric mice fed 12 weeks of HFD (Fig. 2a), indicating that robust diphtheria-toxin-induced Treg ablation (Fig. 5b) had no overt effect on weight gain. Diphtheria toxin treatment attenuated insulin sensitivity (Fig. 5c) and increased fasting insulin (Supplementary Fig. 6) in HFD-fed aMHCII$^{-/-}$ but not WT mice, and similarly increased CD4$^{+}$ ARTs and decreased CD8$^{+}$ ARTs in both genotypes, but had no effect on splenic CD4$^{+}$ and CD8$^{+}$ T cell percentages, despite robust ablation of splenic Tregs (Fig. 5d and Supplementary Fig. 7).

As use of diphtheria toxin ablates total body Tregs, we employed ST2 antibody (Ab) to specifically decrease adipose tissue Tregs[25]. Mice receiving ST2Ab versus IgG control revealed similar body weight and %body fat as those receiving control IgG (Fig. 5e), but had less VAT Tregs and markedly decreased Tregs

expressing ST2 with no differences in splenic Tregs (Fig. 5f). ST2Ab treated mice also had greater HOMA-IR, worse insulin responses, and greater glucose excursions in response to intraperitoneal glucose (Fig. 5g–i). Thus, two approaches to decrease VAT Tregs in aMHCII$^{-/-}$ mice induced insulin resistance, suggesting that maintenance of adipose Treg abundance, independent of splenic Treg homoeostasis, is required for the attenuated adipose inflammation and IR in HFD-fed aMHCII$^{-/-}$ mice.

## Discussion

Relatively little is known about the mechanisms through which adipocytes respond to caloric excess to initiate adipose inflammation. Our previous work uncovered a dialogue between adipocytes and CD4$^{+}$ T cells, whereby an increase in leptin within days of HFD increases ARTs expression of the Th1 marker Tbet and IFNγ, which increases adipocyte MHCII expression[10]. The elevated adipocyte MHCII further enhances Tbet and IFNγ expression in ARTs, leading to an escalating cycle of inflammation. These changes occur without an increase in

macrophage MHCII expression or an increase in ATM abundance. In order to more specifically define the relative contributions of adipocytes and ATMs to obesity-induced CD4$^+$ ART changes, we generated mice with adipocyte-specific MHCII deficiency. We report herein that knockdown of adipocyte MHCII protects mice from diet-induced IR by maintaining adipose Treg abundance and inhibiting pro-inflammatory adipocyte, ART and ATM changes associated with obesity. HFD-fed aMHCII$^{-/-}$ mice are highly resistant to a characteristic HFD-induced decrease in adipose Tregs, but VAT Treg ablation attenuates the improved insulin action of HFD-fed obese aMHCII$^{-/-}$ mice. IFNγ plays a major role in preventing Treg accumulation in obesity, blocking the actions of IL-33 to stimulate Treg differentiation and expression of the IL-33 receptor ST2, and HFD-fed IFNγR1$^{-/-}$ mice are more insulin sensitive than their WT counterparts, which is associated with increased adipose Tregs that also express greater amounts of ST2. Taken together these results indicate that adipocyte APC activity is functionally significant and that maintenance of adipose Treg abundance in HFD-fed aMHCII$^{-/-}$ mice is responsible for their resistance to adipose inflammation and IR during HFD challenge.

We did not expect the significant decrease in H2-Ab1 expression only in eWAT but not in iWAT. Cre expression in iWAT is reportedly lower in iWAT than that in eWAT in adipo-cre mice[26], which appears consistent with our data. In support of these findings, adiponectin expression is higher in visceral fat than in subcutaneous fat in rats[27]; and in humans, secretion of adiponectin by omental adipocytes is reported to be higher than by subcutaneous adipocytes[28]. In addition, the H2-Ab1 gene expression level is much lower in iWAT than eWAT (Supplementary Fig. 1). Thus, in our case adipoq-cre has low efficiency in iWAT to deplete the H2-Ab1 gene, which is relatively lowly expressed particularly in lean mice. Because visceral adipose tissue is the major site that develops chronic inflammation and regulates metabolism in obesity, the decrease of H2-Ab1 gene in visceral adipose tissue effectively changed adipose inflammation and insulin sensitivity in aMHCII$^{-/-}$ mice.

Adipose Tregs represent ∼50% of the CD4$^+$ ART population in lean mice versus 10–15% of CD4$^+$ T cells in lymphoid tissues[19], but their abundance declines to ∼20% during obesity[3,29]. Indeed, in adipose of obese aMHCII$^{-/-}$ mice, Tregs accounted for nearly 50% of the CD4$^+$ T cells consistent with their improved glucose tolerance and insulin sensitivity. Tregs attenuate pro-inflammatory responses to foreign antigens by inhibiting adjacent T cells, blocking both direct effects on the local tissue microenvironment, as well as actions to recruit additional immune cells to promote inflammation. Immune cells play central roles in maintaining metabolic homoeostasis in lean adipose tissue[30]. In particular, adipose Tregs are essential for attenuating ART and ATM inflammatory cascades leading to adipocyte dysfunction and systemic IR. However, adipose Treg decreases associated with caloric excess disrupt this anti-inflammatory balance and correlate with increased adipose inflammation and IR during weight gain[3–5,31]. Very little is known about what causes the downregulation of adipose Treg abundance during excess weight gain; our results indicate that adipocyte MHCII is an important determinant in this process.

Chronic inflammation itself does not appear to inhibit Treg homoeostasis, as Tregs remain elevated during autoimmune reactions and chronic infections at other anatomical sites[32]. Adipose Tregs are positively regulated by IL-2 produced by iNKT cells and IL-33 from adipocytes and endothelial cells[20,33]. Reduced iNKT cell prevalence in adipose tissue of obese mice may decrease adipose Treg abundance[33]. IL-33 promotes adipose-specific Treg differentiation and proliferation, but

increased human and mouse adipose tissue IL-33 expression during obesity is not sufficient to prevent adipose Treg declines. This obesity-linked IL-33 resistance may be overcome by IL-33 treatment, which increases the number and ST2$^+$ percentage of adipose Tregs in HFD-fed mice[22]. We found that adipose IL-33 expression does not differ by genotype, but that aMHCII$^{-/-}$ mice have a higher percentage of ST2$^+$ Tregs than WT mice and thus should be more sensitive to IL-33 effects to promote adipose Treg proliferation. Attenuated adipocyte MHCII expression, thus, participates in preservation of IL-33 sensitivity to maintain adipose Treg abundance, which is lost with blockade of ST2.

Reduced IFNγ expression can explain many of the phenotypic improvements observed in our HFD-fed aMHCII$^{-/-}$ mice. IFNγ-deficient mice exhibit less metabolic dysfunction when fed HFD[34], IFNγ-deficient Tbet null mice have substantially more adipose Tregs than WT mice[7], and we and others have found that IFNγ inhibits Treg differentiation and accumulation[20,35,36]. Moreover, HFD-fed IFNγR1$^{-/-}$ mice exhibited insulin sensitivity and adipose Treg phenotypes similar to those of aMHCII$^{-/-}$ mice, strongly suggesting that IFNγ drives the adverse metabolic and immune changes observed in HFD-fed WT mice. Adipose IFNγ expression is increased in obese humans and mice[3,4,10] and may reduce adipose Treg abundance by attenuating IL-33-driven Treg accumulation[20]. We found that IFNγ directly inhibits the ability of IL-33 to stimulate adipose Treg ST2 expression, as well as proliferation, but has no effect on the anti-inflammatory activity of these cells. Thus, the drop in IFNγ in aMHCII$^{-/-}$ adipose is a likely candidate contributing to the increase in Tregs in this model.

ATMs are a major adipose tissue component in lean and obese mice and have long been proposed to act as the APCs that regulate CD4$^+$ ART-driven adipose inflammation. Correspondingly, mice with myeloid-selective MHCII-deficiency (mMHCII$^{-/-}$) are reported to exhibit reduced adipose inflammation and IR when fed HFD[37]. HFD-fed aMHCII$^{-/-}$ and mMHCII$^{-/-}$ mice, however, exhibit significant differences in their ART and metabolic phenotypes. Adipose Tregs are decreased in HFD-fed mMHCII$^{-/-}$ versus WT mice, but increased in HFD-fed aMHCII$^{-/-}$ versus WT mice. We also observed an increase in the ratio of anti-inflammatory Tregs to pro-inflammatory Tconvs in adipose tissue from aMHCII$^{-/-}$ mice, but there was no apparent difference in this ratio in HFD-fed mMHCII$^{-/-}$ mice to explain their reported reduced IFNγ expression and improved insulin sensitivity. Non-adipose tissue myeloid MHCII-deficiency may also contribute to HFD-fed mMHCII$^{-/-}$ mouse phenotypes, as myeloid-driven inflammatory processes also regulate HFD-induced systemic IR through actions on additional metabolic tissues, including pancreatic islets, liver and skeletal muscle. MHCII-deficiency has also been reported to alter macrophage innate immune responses via an effect to attenuate TLR signalling[38], although similar results were not observed by Cho et al.[33] Thus, adipose-specific versus systemic effects to explain the mMHCII$^{-/-}$ phenotype are unclear.

Chronic exposure to high levels of saturated free fatty acids, glucose and insulin during obesity is reported to stimulate a 'metabolically-activated' macrophage phenotype more characteristic of the ATM response to HFD diet than the 'classically-activated' M1 macrophage response to microbial antigens[18]. Palmitate is considered a major driver of the metabolically activated phenotype. Palmitate is the most common saturated fatty acid in the body and is known to activate inflammation in macrophages through toll-like receptors that can promote an M1 phenotype. Multiple markers of the metabolically activated phenotype (CD36, ABCA1 and Plin2), as well as some M1 markers, are reduced in HFD-fed aMHCII$^{-/-}$

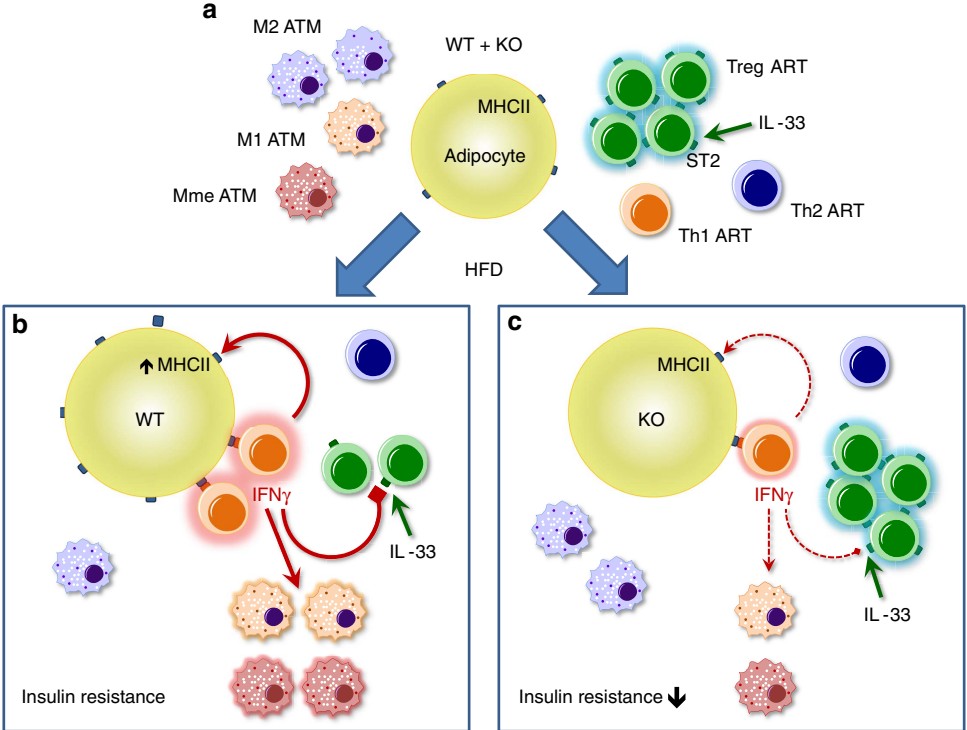

**Figure 6 | Adipocyte MHCII can regulate HFD-induced adipose inflammation via IFNγ.** (**a**) Adipose tissue of chow-fed lean mice is characterized by low-level adipocyte MHCII expression and anti-inflammatory balance of Treg + Th2 to Th1 ARTs and M2 to M1$^+$ metabolically-activated (Mme) ATMs. (**b**) HFD-induced obesity robustly increases Th1 IFNγ that can (1) increase adipocyte MHCII to further promote Th1 activation and (2) inhibit adipose Treg ST2 expression to reduce IL-33-driven proliferation to directly and indirectly stimulate M1 and Mme ATM polarization. (**c**) HFD-fed aMHCII KO mice demonstrate attenuated ART IFNγ expression, with reductions in all these pro-inflammatory phenotypes.

versus WT mice. Further studies are needed to determine the cause of these changes. In 3T3L-1 adipocytes there was no difference in rates of lipolysis when cells were treated with a siRNA for CIITA or scrambled control[10], and there was no difference is expression of lipolysis genes in aMHCII$^{-/-}$ versus WT adipocytes (data not shown). In addition, decreased IFNγ expression in ARTs of aMHCII$^{-/-}$ mice likely contributes to the decreased M1 markers.

Results presented herein clearly indicate that adipocyte MHCII plays a central role in inducing CD4$^+$ ART activation and IFNγ production in HFD-fed mice to decrease adipose Treg homoeostasis and thereby promote adipose inflammation and systemic IR (Fig. 6). Interrupting this dialogue by attenuating adipocyte MHCII expression or decreasing IFNγ signalling attenuates adipose inflammation and preserves adipose Tregs and systemic insulin sensitivity, suggesting new therapeutic targets for interventions to attenuate the negative phenotypes associated with excess weight gain.

## Methods

**Mouse studies.** C57BL/6J (JAX#000664), and IFNγR KO (JAX#003288), B6.H2-Ab1$^{flox/flox}$ (JAX#013181), B6.Adipoq-Cre (JAX#010803), B6.Foxp3-DTR-EGFP (JAX#016958) and OTII (B6.Cg-Tg(TcraTcrb)) (JAX#004194) mice were purchased from the Jackson Laboratory. B6.H2-Ab1$^{flox/flox}$ mice were crossed with B6.Adipoq-Cre to create adipocyte-specific MHCII knockout mice (aMHCII$^{-/-}$). B6.H2-Ab1$^{flox/flox}$ (WT) mice and aMHCII$^{-/-}$ mice were crossed to create experimental mouse litters. Male mice were group- housed under a 12 h light/dark cycle, and fed standard diet (14% kcal% fat; 8904, Harlan Teklad) until 8–10 weeks-of-age then switched to high-fat diet (HFD; 60% kcal% fat; D12492, Research Diets, New Brunswick, NJ), to induce weight gain. These bone marrow transplant (BMT) mice were engrafted with bone marrow from 10-week old male B6.Foxp3-DTR-EGFP mice[39] and switched to HFD at 4 weeks post-BMT. After 10 weeks HFD, mice were IP-injected twice-weekly for 4 weeks with PBS or diphtheria toxin (Roche, 6.25 ng g$^{-1}$ body weight) while fed HFD[40]. Mice were weighed, analysed for body composition and insulin tolerance, as previously

reported[10]. GTTs were performed on overnight-fasted, non-anesthetized mice, using tail vein blood samples obtained 0, 15, 30, 45, 60 and 120 min after intraperitoneal glucose injection (1 g kg$^{-1}$ body weight). aMHCII$^{-/-}$ mice that had been on HFD for 12 weeks were IP injected with 300 μg Mouse ST2/IL-33R antibody (R&D Systems, Minneapolis, MN) or Rat IgG2b (Bio X Cell, West Lebanon, NG) every other day for four injections. ITTs and GTTs were performed at the end of antibody treatment and the animals sacrificed. All animal procedures were conducted in specific-pathogen-free facilities at The Methodist Hospital Research Institute and Ohio State University. Ethical approval of the work was granted by both the Houston Methodist Animal Care Committee and The Ohio State University Institutional Animal Care and Use Committee, in accordance with institutional animal care and use committee guidelines.

**Adipose tissue histology.** Mouse epididymal white adipose tissue samples were fixed, sectioned and stained as previously described and imaged to analyse adipose phenotypes[10]. Crown-like structure frequency was determined as the average number of adipocytes surrounded by macrophages per unit area, as determined from 25 images per mouse[41]. Adipocyte diameter was measured by counting 100 adipocytes per mouse using Nikon Elements AR imaging software[42].

**Adipose cell isolation and gene expression analyses.** Adipocyte and adipose SVF, ART and ATM fractions were isolated from epididymal VAT[10]. Briefly, adipocytes were incubated with biotinylated CD45 antibody (eBiosciences, 13-0451-85, 1:400 dilution) to remove leukocyte contamination and the purified CD45 negative adipocytes were used for the gene expression analysis and cell co-culture. ATMs and ARTs were isolated from SVF with biotinylated F4/80 (eBiosciences, 13-4801-85, 1:200 dilution) and CD3e (eBiosciences, 13-0031-85, 1:100 dilution) antibodies, respectively. BMDMs and PMs were generated[39]. DCs were isolated from spleens of WT and aMHCII$^{-/-}$ mice by using Miltenyi Pan Dendritic Cell Isolation Kit. Cell and tissue RNA samples were isolated and analysed, as previously described[10].

**Western blotting analyses.** To examine the H2-Ab1 protein expression in aMHCII$^{-/-}$ mice, adipocytes and SVFs were isolated from HFD-fed WT and aMHCII$^{-/-}$ mice. To check the insulin signalling, mice were fasted for 6 h and intraperitoneally injected with insulin (4 U per kg body weight) or PBS. Skeletal muscle and eWAT were harvested 15 min after injection. Cells and tissues were lysed in lysis buffer containing 50 mM Tris (pH 7.5), 1% Nonidet P-40,

0.1% sodium dodecyl sulfate (SDS), 150 mM NaCl, 0.5% sodium deoxycholate, PhosSTOP (Roche), and protease inhibitors (Roche). Protein separation, immunoblotting and visualization were performed as described previously[43]. H2-Ab1 antibody was purchased from Abcam (Cat#ab63567, 1:250 dilution). Phospho-AKT (Cat#4060, 1:1,000 dilution) and AKT (Cat#9272, 1:1,000 dilution) antibodies were purchased from Cell Signaling Technology. Full-size immunoblots are provided in Supplementary Fig. 9.

**Flow cytometry analyses.** Fluorochrome-conjugated antibodies directed against the following mouse antigens were used for analysis by flow cytometry. BioLegend: CD4 (Cat#100509, 1:250 dilution), F4/80 (Cat#123113, 1:300 dilution), CD206 (Cat#141712, 1:100 dilution), CD11c (Cat#117322, 1:100 dilution), CD11b (Cat#101227, 1:250 dilution), ST2 (Cat#145312, 1:100 dilution); eBiosciences: CD45 (Cat#12-0481-82, 1:667 dilution), CD36 (Cat#51-0361-80, 1:200 dilution), Foxp3 (Cat#17-5773-80, 1:20 dilution), CD3 (Cat#48-0032-82, 1:100 dilution), CD8a (Cat#25-0081-82, 1:350 dilution). Cells were incubated with Zombie dye (BioLegend) at room temperature for 15 min, then incubated with CD16/32 antibody (eBioscience, 14-0161-85, 1:50 dilution) for another 15 min and stained with detection antibodies for 10 min at room temperature. Stained cells were washed twice in PBS and fixed in 1% formaldehyde before analysis. Intracellular staining was performed using eBioscience Foxp3 staining kit according to the manufacturer's protocol. Analyses were performed on BD LSRII Flow Cytometer (BD Biosciences) and analysed with FlowJo software (Treestar). Cells were first gated for singlets (FSC-H versus FSC-A) and live leukocytes (Zombie$^-$CD45$^+$), which were then gated for macrophages (F4/80$^+$CD11b$^+$) or T cells (CD3$^+$). Macrophages were analysed for expression of CD11c, CD206 and CD36. T cells were analysed for expression of CD4 and CD8. CD4$^+$ T cells were analysed for the expression of IFNγ or Foxp3, and Foxp3$^+$ T cells were analysed for ST2 expression.

**Adipocyte-T cell co-culture studies.** Co-cultures of primary mouse adipocytes and CD4$^+$ T cells were performed, as previously described in ref. 10. Adipocytes of WT and aMHCII$^{-/-}$ mice fed 12 weeks of HFD were cultured with naïve CD4$^+$ T cells isolated from ovalbumin-reactive mice (OTII) and cultured with low-dose ovalbumin, IL-2, and TGFβ (ref. 44) in the presence of indicated amounts of recombinant IFNγ or conditioned media (50%) from adipocyte APC activation cultures supplemented with 10 μg ml$^{-1}$ IFNγ blocking antibody (eBioscience, BMS182) or non-specific isotype control antibody (Bio X Cell, BE0089)[45]. Treg cells were measured by flow cytometry analysis.

**Adipose Treg proliferation and function studies.** For the Treg proliferation study, SVF and spleen cells were isolated from 20-30 weeks old male mice and lymphocytes were isolated on Histopaque (Sigma) gradients. Lymphocytes (1 × 10$^5$) were cultured in IMDM with 10% FBS in U bottom 96 well plate and treated with or without IL-33 (10 ng ml$^{-1}$) in the presence or absence of IFNγ (5 ng ml$^{-1}$) for 3–4 days. Treg cell number and ST2$^+$ Treg percentages were measured by flow cytometry. To assess Treg function, adipose CD4$^+$CD25$^+$ Tregs and spleen CD4$^+$CD25$^+$ Tregs CD4$^+$CD25$^-$ effector T cells were sorted from epididymal adipose and spleen of 20-30 weeks old male mice. Spleen CD4$^+$CD25$^-$ effector T cells (1 × 10$^4$) were cultured in 96-well plates in 200 μl of RPMI supplemented with 0.5 μg ml$^{-1}$ of anti-CD3 mAb and irradiated antigen-presenting cells (2 × 10$^4$) with Treg cells were titrated in at the indicated ratios. Cultures were performed in triplicate, incubated with or without 5 ng ml$^{-1}$ IFNγ for 4 days, and pulsed with $^3$H-thymidine (1 μCi per ml) for the last 16 h of each experiment. Proliferation values were normalized to those of effector T cells cultured without Tregs or supplemental IFNγ.

**Statistical analyses.** Mean ± s.e. data and sample sizes are reported in figure legends. Prism 6.0 software (Graphpad, San Diego, CA) was used for all statistical analyses. Differences between groups were analysed using Mann-Whitney U-tests or Student's or Welch's T-tests as indicated by data normality, variation and statistical power.

**Data availability.** All data are available within the article (as figure source data or Supplementary Information Files) and/or from the authors on request.

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

## Acknowledgements

This work was supported by a generous gift from Ms V. Patel to W.A.H.; an AHA grant 14SDG18970097, an ADA grant 1-17-IBS-179 and Innovation-driven Project of Central South University (No. 2017CX011) to T.D.; and a NIH grant R01AI106200 to X.C.L.; and CA210087, CA95426 and CA68458 to M.C.

## Author contributions

T.D., C.J.L. and W.A.H. conceived and designed the experiments. T.D. performed most of the experiments. L.M., V.W. and R.Y. contributed to animal handling and phenotyping. J.L. and A.B. provided technical supports in key experiments. X.X., Y.D. and S.B. performed flow cytometry analysis. X.C.L., M.C. and D.B. provided helpful discussions, T.D., C.J.L., D.D. and W.A.H. analysed the data and wrote the manuscript.

## Additional information

**Competing interests:** The authors declare no competing financial interests.

