## [Peer Review File · Nature Communications]

Editorial Note: Parts of this peer review file have been redacted as indicated to maintain confidentiality of communication between the authors and a collaborator.

Reviewers' Comments:

Reviewer #1 (Remarks to the Author)

1. Figure 5. The results in this figure require some further explanation. The body weights in Figure 5 are about the same as in Figure 2A, as the Authors note. However, BMT itself usually attenuates body weight gain on HFD, so can the Authors comment on why the HFD BWs are the same in non-chimeric (Figure 2A) and chimeric (Figure 5A) mice. The BMT experiments would be improved with a more thorough metabolic analysis such as GTTs, GSIS, and some measures of in vivo insulin action such as AKT phosphorylation. In addition, Figure S5 shows that basal insulin levels rise with DT injections in the KO, but there is no change in glucose levels. What is the mechanism of the decrease in insulin secretion in the absence of a change in glucose? Was there an increase in some other secretagogue, i.e. GLP1, or some other mechanism.
2. In Figure 1, why was there no decrease in H2AB1 expression in IWAT and only in EWAT, since the KO is adipocyte selective.
3. Why is the degree of KO in adipocytes only about 50%?
4. In Figure 1C it appears as if all the data were normalized to WT which is 1.0. What is the relative comparison of H2AB1 mRNA between EWAT and IWAT, independent of this method of normalization?
5. As stated for Figure 2, the Authors have studied these mice at 6 weeks and 12 weeks. The 12-week data is in Figure 2 and the 6-week data is in Figure S1. However, some of the panels are not comparable measurements. For example, what does the GTT show at 6 weeks? Can the Authors provide a comparison of TH1 subtype, TREG, and macrophage content in the 6-week HFD vs. the 12 week HFD WT and KO groups.
6. What is the mechanism of the decreased insulin levels at 6 and 12 weeks in Figure 2D without any change in glucose levels. It is clear that the KOs are more sensitive, but there must be some signaling mechanism to the beta cell leading to the decreased insulin secretion.
7. In their descriptions, the Authors tend to intermix the concepts of differential cell expression vs. mRNA expression. They need to be more specific when they are describing mRNA differences that might be ascribed to a particular cell type vs. actual differences in TREGs or macrophages.
8. GTTs and plasma insulin levels should be provided for the studies in Figure S4.
9. If I am interpreting Figure 5C correctly, they are showing that depletion of TREGs with DT injection does not change the ITTs in the WT mice. This is surprising given the general hypothesis of others as well as the current Authors that TREGs promote insulin sensitivity. Please explain.
10. In Figure 5C, are the WT PBS and KO PBS different with respect to insulin sensitivity?
11. Since TREGs are dependent on MHCII, what is the mechanism for the increased TREGs in the KO mice?
12. It would be helpful that they provided some data on tissue insulin sensitivity, such as insulin stimulated AKT phosphorylation in their models to see how well this correlates with the ITTs.
13. In Figure 2, there are differences in ITTs and GTTs between the WT and KO but differences don't seem to exist in Figure S4 panel B. Please explain.
14. Figure 6 appears to indicate the hypothesis that lipolysis is decreased in MHCII KO adipocytes. Do the authors have experimental evidence to support this?
15. The comments about TREG PPAR γ in the discussion are a bit confusing given the previous literature on TREG PPAR γ . Please expand and explain.

Reviewer #2 (Remarks to the Author)

Some points for the authors to consider:

1. The authors characterized the T cell populations in the HFD mice, but it doesn't appear the same careful analysis was performed in CHO fed mice. This would be of interest: from what initial populations does the HFD cause changes?

2. Figure 2D shows ITT data, plus glucose and insulin levels and calculates HOMAR-IR, but these values are equivocal in terms of underlying mechanisms. Thus the text that describes "insulin resistance" is not warranted because the differences are small and would have to be confirmed by full hyperinsulinemic clamp measurements on these animals. The authors need to perform such measurements to confirm their initial conclusions. For Figure 5, it would also be good to include clamp data since this is a key aspect of the paper.

3. Figure 3. The small increases in mRNA for adiponectin and PPAR γ shown here are not compelling in terms of mechanistic information. It would be important to measure actual protein levels of these key entities, not just message. In the case of PPAR γ , it's the activity that is key, not the total protein level, so to claim PPAR γ is more active is a stretch. This would at least have to be confirmed by measuring PPAR γ -sensitive gene sets (e.g., metabolic genes) to show they are increased as expected.

Reviewer 1: Thank you for your very helpful and thorough comments. We believe they helped us to substantially improve the manuscript.

1. Figure 5. The results in this figure require some further explanation. The body weights in Figure 5 are about the same as in Figure 2A, as the Authors note. However, BMT itself usually attenuates body weight gain on HFD, so can the Authors comment on why the HFD BWs are the same in non-chimeric (Figure 2A) and chimeric (Figure 5A) mice. The mice in Figure 2A were fed HFD for 12 weeks. The bone marrow transplant (BMT) mice in Figure 5A were allowed to recover for 4 weeks and then administered HFD for 14 weeks. Usually the non-chimeric mice gain more weight than chimeric mice on HFD; however, the BMT mice were 6 weeks older and were fed HFD 2 more weeks than the non-chimeric mice. This difference allowed both groups to have similar body weight at the end points of these experiments. The BMT experiments would be improved with a more thorough metabolic analysis such as GTTs, GSIS, and some measures of in vivo insulin action such as AKT phosphorylation. The aim of the experiment shown in Figure 5 was to determine if depletion of the adipose Treg population observed in the HFD-fed aMHCII KO mice was sufficient to alter their insulin sensitivity phenotype to resemble that of the HFD-fed WT mice. We used ITTs to examine insulin sensitivity of these mice, since this assay directly measures insulin-induced glucose disposal. This experiment was intended only to validate the contribution of the increased adipose Treg levels in HFD-fed aMHCII KO mice to insulin sensitivity, since it has already been established that adipose Treg abundance can regulate insulin sensitivity (Feuerer et al, 2009, Ilan et al., 2010; Winer et al., 2009). Loss of Tregs in HFD-fed aMHCII^{-/-} mice worsens insulin sensitivity to that of HFD-fed WT mice, suggesting a role of Tregs in maintaining insulin sensitivity in aMHCII^{-/-} mice.

Rather than repeat this experiment to obtain more metabolic data, we administered ST2 antibody (Ab) to aMHCII^{-/-} mice in order to more precisely target visceral adipose (Bapat, Nature, 2015) and not peripheral Tregs (Fig. 5E-H). We found that the ST2 Ab decreased Treg in visceral adipose tissue (VAT) but not in spleen, and worsened ITT and ipGTT responses compared to IgG controls. Thus, two approaches to decrease Tregs in aMHCII^{-/-} mice, one specifically targeting VAT Tregs, worsened insulin sensitivity, suggesting that adipose tissue Tregs are responsible for the improved insulin sensitivity in aMHCII^{-/-} mice. In addition, Figure S5 shows that basal insulin levels rise with DT injections in the KO, but there is no change in glucose levels. What is the mechanism of the decrease in insulin secretion in the absence of a change in glucose? The fasting serum insulin level is less in HFD-fed aMHCII^{-/-} compared to WT mice despite similar fasting glucose levels because the aMHCII^{-/-} mice are more insulin sensitive and require less insulin to maintain glucose homeostasis. This mechanism is defined in Bergman's 2006 Banting lecture (Diabetes, 2007) where he demonstrated that insulin resistance could be compensated by increased insulin secretion. HFD-fed normal dogs developed severe insulin resistance and hyperinsulinemia *without* an increase in either fasting or 24 hour glucose excursions. This physiology also explains the fasting glucose/insulin relationships following DT administration.

2. In Figure 1, why was there no decrease in H2AB1 expression in IWAT and only in EWAT, since the KO is adipocyte selective. Cre expression in IWAT is reportedly lower in IWAT than that in EWAT in adipo-cre mice (Eguchi J. et al. Rosen, Cell Metab, 2011, Fig. S3b), which appears consistent with our data. In support of these findings, adiponectin expression is higher in visceral fat than in subcutaneous fat in rats (Atzmon G, Horm Metab Res, 2002); and in humans, secretion of adiponectin by omental adipocytes is reported to be higher than by subcutaneous adipocytes (Motoshima H, JCEM, 2002). In addition, the H2AB1 gene expression level is much lower in IWAT than EWAT (Fig. S1). Thus, in our case adipoq-cre has low efficiency in IWAT to deplete the H2AB1 gene, which is relatively lowly expressed particularly in lean mice.

3. Why is the degree of KO in adipocytes only about 50%? We agree that the adipoq-cre driven deletion of H2Ab1 mRNA adipocytes from chow fed aMHCII^{-/-} vs. WT mice is not robust, likely because the basal level of H2Ab1 in lean mice is not high. However, differences in H2Ab1 deletion are more prominent in adipocytes from HFD-fed aMHCII^{-/-} vs. WT mice because high fat feeding increases adipocyte H2Ab1 expression in WT but not in aMHCII^{-/-} H2Ab1 mice (please see H2Ab1 protein blot, Fig. 1E). Importantly, we also have functional data to support decreased adipocyte MHCII in aMHCII^{-/-} mice (Fig. 1F).

4. In Figure 1C it appears as if all the data were normalized to WT which is 1.0. What is the relative comparison of H2AB1 mRNA between EWAT and IWAT, independent of this method of normalization H2Ab1 expression is higher in eWAT than iWAT; these data are now added in new Figure S1.

5. As stated for Figure 2, the Authors have studied these mice at 6 weeks and 12 weeks. The 12-week data is in Figure 2 and the 6-week data is in Figure S1. However, some of the panels are not comparable measurements. For example, what does the GTT show at 6 weeks? This data is now included in Figure S2C. Can the Authors provide a comparison of TH1 subtype, TREG, and macrophage content in the 6-week HFD vs. the 12 week HFD WT and KO groups. We examined the Th1, Treg, and macrophages in VAT from the 6-week HFD WT and KO mice. The percentage of Treg and ST2⁺ Treg are higher in KO mice (Fig. S2H). However, the Th1 and macrophages show no difference between WT and KO (data not shown). These data are now added to the text of the Results section.

6. What is the mechanism of the decreased insulin levels at 6 and 12 weeks in Figure 2D without any change in glucose levels. It is clear that the KOs are more sensitive, but there must be some signaling mechanism to the beta cell leading to the decreased insulin secretion. Please see above comment relative to Bergman's Banting lecture.

7. In their descriptions, the Authors tend to intermix the concepts of differential cell expression vs. mRNA expression. They need to be more specific when they are describing mRNA differences that might be ascribed to a particular cell type vs. actual differences in TREGs or macrophages. We apologize for the confusion. We isolated ATM and ARTs fractions and measured gene expression of biomarker genes, but we also performed flow studies that measured number/percent cells containing specific marker proteins which is a better indicator of changes in Tregs or specific types of macrophages.

8. GTTs and plasma insulin levels should be provided for the studies in Figure S4. We provided fasting insulin/glucose data for DT treated mice in Figure S6. We have now added ST2 experiments for which fasting insulin and glucose, HOMA-IR, ITT and ipGTT are provided (Fig 5 E-I).

9. If I am interpreting Figure 5C correctly, they are showing that depletion of TREGs with DT injection does not change the ITTs in the WT mice. This is surprising given the general hypothesis of others as well as the current Authors that TREGs promote insulin sensitivity. Please explain. Systemic Treg ablation by DT treatment has little effect on insulin resistance in HFD-fed WT mice, likely because they already exhibit diet-induced adipose Treg deficiency. This observation does not contradict our hypothesis, because it is unclear that there is a continuous relationship between Treg abundance and inflammation/insulin resistance. It is more likely that there is a discontinuous relationship, since there may be a threshold after which

reducing Treg abundance has diminishing returns to induce more adipose inflammation and insulin resistance.

10. In Figure 5C, are the WT PBS and KO PBS different with respect to insulin sensitivity? Yes, this data is shown below. There is no difference between WT and KO in DT groups.

11. Since TREGs are dependent on MHCII, what is the mechanism for the increased TREGs in the KO mice? Two mechanisms may explain the increased Tregs in aMHCII^{-/-} mice. The first involves decreased adipose IFN γ in the KO mice (Fig. 3B). CD4⁺ Th1 cells and CD8⁺ T cells produce IFN γ and markers for these cells are also decreased in this figure. In addition, we demonstrate that naïve OTII T cells incubated with adipocytes of KO mice generate less IFN γ than that T cells incubated with WT adipocytes. Several studies (Ajithkumar v. Nature Immunology, 2015; Mathis, cell Metabolism 2015) demonstrated that IL-33 plays a key role in proliferation of adipose Tregs. Our current study shows that IFN γ dose-dependently inhibits Treg differentiation and IL-33 receptor expression on Tregs, and IFN γ -R KO mice are more insulin resistant with less visceral adipose Tregs than WT mice (Fig. 4). These data suggest IFN γ is a major regulator of Treg abundance by inhibiting IL-33 signaling. The second mechanism involves non-adipocyte MHCII, primarily macrophages and B cells. We agree that some MHCII is necessary for Treg differentiation and although adipocyte MHCII is genetically reduced, adipose tissue macrophages still express MHCII. Indeed, loss of macrophage MHCII leads to markedly decreased adipose Tregs (Lumeng C, Cell Reports, 2015).

12. It would be helpful that they provided some data on tissue insulin sensitivity, such as insulin stimulated AKT phosphorylation in their models to see how well this correlates with the ITTs. These data are now added to Figure 2E.

13. In Figure 2, there are differences in ITTs and GTTs between the WT and KO but differences don't seem to exist in Figure S4 panel B. Please explain. We agree there are no differences in Figure S4 B. We performed these ITTs after the bone marrow transplant when the mice had only been on HFD for 4 weeks, prior to the appearance of differences in insulin sensitivity. We agree these data are confusing and should be removed.

14. Figure 6 appears to indicate the hypothesis that lipolysis is decreased in MHCII KO adipocytes. Do the authors have experimental evidence to support this? We do not have evidence to support this since we found no difference in expression of lipolytic genes in adipocytes of HFD-fed WT and KO mice. Lipolysis has been removed from Figure 6.

15. The comments about TREG PPAR γ in the discussion are a bit confusing given the previous literature on TREG PPAR γ . Please expand and explain. Please see Reviewer 2's comment 3

below. Because of the small difference in adipocyte PPAR γ gene expression and the lack of differences in ARTs PPAR γ expression (data not shown) we removed PPAR γ comments from the Discussion. We agree they were confusing.

Reviewer 2: Thank you for your excellent review.

1. The authors characterized the T cell populations in the HFD mice, but it doesn't appear the same careful analysis was performed in CHO fed mice. This would be of interest: from what initial populations does the HFD cause changes? T cell flow data has now been added for the chow-fed mice. Please see Figure 1I.

2. Figure 2D shows ITT data, plus glucose and insulin levels and calculates HOMAR-IR, but these values are equivocal in terms of underlying mechanisms. Thus the text that describes "insulin resistance" is not warranted because the differences are small and would have to be confirmed by full hyperinsulinemic clamp measurements on these animals. The authors need to perform such measurements to confirm their initial conclusions. For Figure 5, it would also be good to include clamp data since this is a key aspect of the paper. We have added western blots of insulin-induced pAKT in skeletal muscle and adipose tissue and see differences between aMHCII^{-/-} and WT mice (Fig. 2E). Although clamps may be interesting and support the HOMA-IR, ITT, ipGTT, and western blot data, the cost is prohibitive. [redacted]

3. Figure 3. The small increases in mRNA for adiponectin and PPAR γ shown here are not compelling in terms of mechanistic information. It would be important to measure actual protein levels of these key entities, not just message. In the case of PPAR γ , it's the activity that is key, not the total protein level, so to claim PPAR γ is more active is a stretch. This would at least have to be confirmed by measuring PPAR γ -sensitive gene sets (e.g., metabolic genes) to show they are increased as expected. We agree. We performed protein blots on adipose tissue and did not see differences in adiponectin and PPAR γ proteins. Thus, it is unlikely that adiponectin or PPAR γ is responsible for the insulin sensitive phenotype of aMHCII^{-/-} mice. We eliminated the PPAR γ comments from the Discussion.

Reviewers' Comments:

Reviewer #1:
None

Reviewer #2:
None